# Emerging Head and Neck Tumor Targeting Contrast Agents for the Purpose of CT, MRI, and Multimodal Diagnostic Imaging: A Molecular Review

**DOI:** 10.3390/diagnostics14151666

**Published:** 2024-08-01

**Authors:** Jonathan M. Neilio, Daniel T. Ginat

**Affiliations:** 1Chicago Medical School, Rosalind Franklin University, North Chicago, IL 60064, USA; jonathan.neilio@my.rfums.org; 2Department of Radiology, University of Chicago, Chicago, IL 60637, USA

**Keywords:** MRI, CT, tumor, contrast, targeting, molecular, head and neck

## Abstract

Background. The diagnosis and treatment of head and neck tumors present significant challenges due to their infiltrative nature and diagnostic hindrances such as the blood–brain barrier. The intricate anatomy of the head and neck region also complicates the clear identification of tumor boundaries and assessment of tumor characteristics. Aim. This review aims to explore the efficacy of molecular imaging techniques that employ targeted contrast agents in head and neck cancer imaging. Head and neck cancer imaging benefits significantly from the combined advantages of CT and MRI. CT excels in providing swift, high-contrast images, enabling the accurate localization of tumors, while MRI offers superior soft tissue resolution, contributing to the detailed evaluation of tumor morphology in this region of the body. Many of these novel contrast agents have integration of dual-modal, triple-modal, or even dual-tissue targeting imaging, which have expanded the horizons of molecular imaging. Emerging contrast agents for the purpose of MRI and CT also include the widely used standards in imaging such as gadolinium and iodine-based agents, respectively, but with peptide, polypeptide, or polymeric functionalizations. Relevance for patients. For patients, the development and use of these targeted contrast agents have potentially significant implications. They benefit from the enhanced accuracy of tumor detection and characterization, which are critical for effective treatment planning. Additionally, these agents offer improved imaging contrast with the added benefit of reduced toxicity and bioaccumulation. The summarization of preclinical nanoparticle research in this review serves as a valuable resource for scientists and students working towards advancing tumor diagnosis and treatment with targeted contrast agents.

## 1. Introduction

Contrast agents are often administered with CT and MRI scans to help delineate lesions throughout the body. Most MRI contrast agents are gadolinium-based and display a high t_1_ signal, and most CT contrast agents are iodine based, which are hyperattenuating [1,2]. While the use of contrast agents is intended to increase the sensitivity of detecting lesions, they are not necessarily highly specific for particular abnormalities. Hence, there is an impetus for developing contrast agents that can target specific lesions. Peptides and small molecules have shown significant promise for clinical translation due to their high specificity, ease of synthesis, and low toxicity [3]. These agents can be engineered to target specific cancer markers, enhancing the accuracy and effectiveness of diagnostic imaging [3]. In addition to the development of tissue targeting capabilities for contrast agents, novel nanoparticles for cancer imaging potentially pose an advantage over traditional agents due to their superior contrast resolution and the potential for multifunctional applications [4,5]. Although not yet available for clinical use, this article reviews the emerging research and development of several novel tumor-targeting contrast agents.

In the intricate field of head and neck cancer diagnostics, the emergence of metallic nanoparticles (NPs) as contrast agents in CT imaging marks a significant advancement. These novel NPs, in addition to their added tissue-targeting capabilities, present a paradigm shift in how early-stage tumors in the head and neck region are detected and treated. The specific physicochemical properties of these NPs, such as high atomic numbers and superior X-ray absorption, render them more effective than traditional iodine-based agents in enhancing contrast in CT imaging [4].

Gold, silver, bismuth, and other metallic NPs are particularly noteworthy for their enhanced contrast properties [4]. Gold nanoparticles (AuNPs), renowned for their biocompatibility and ease of functionalization, can be engineered to target specific tumor tissues in the head and neck, crucial for early and accurate diagnosis [4]. However, there remains a need for further research into their potential long-term accumulation and toxicity. Copper-based nanoparticles (CuNPs) offer significant contrast enhancement with relatively lower toxicity, but their lower atomic number may slightly reduce their effectiveness in tissue targeting compared to gold or bismuth NPs [4]. Silver nanoparticles (AgNPs), with their unique antibacterial properties, are beneficial post-surgery but raise concerns regarding cytotoxicity and accumulation in the body [4]. Bismuth-based nanoparticles (BiNPs) stand out for their high atomic number leading to superior contrast and lower toxicity, though their stability and potential organ accumulation merit further study [4].

Newer agents can also take advantage of the recent developments in spectral photon-counting computed tomography (SPCCT) imaging that have been significantly bolstered by the utilization of high-atomic-number NPs, offering enhanced contrast and detailed material differentiation crucial for precise medical diagnostics. Studies on NPs such as gold, hafnium, and ytterbium have demonstrated their potential in improving the accuracy of cancer and cardiovascular disease diagnosis, while also highlighting their therapeutic applications [6]. However, the research underscores the need for further development in NP biocompatibility and safety.

The functionalization of these NPs with moieties, like polymers (peptides, proteins, nucleotides, etc.), to target specific cancer markers is particularly advantageous in the heterogeneous landscape of head and neck cancers. This targeted approach ensures that the contrast agents accumulate more in tumor tissues, enhancing the clarity and specificity of CT imaging. Some NPs, like gadolinium and gold, even offer multimodal imaging capabilities, combining the benefits of CT with those of magnetic resonance imaging (MRI) [4]. Despite these promising tissue targeting qualities, the clinical application of metallic NPs faces challenges, primarily concerning their biocompatibility and potential toxicity, especially in sensitive areas like the head and neck. Moreover, the diversity of head and neck cancers may affect the uniformity of NP uptake, impacting imaging efficacy. The journey from laboratory research to clinical practice also involves overcoming regulatory hurdles and rigorous testing processes.

The recent advancements in MRI for head and neck cancer diagnostics have been significantly influenced by the development of various novel NPs as well. These NPs, including iron oxide, manganese (Mn)-based (MnNPs), ^19^F, and CuNPs, present a marked improvement over traditional contrast agents in terms of imaging capabilities [5]. They potentially offer enhanced specificity and sensitivity, crucial for precise tumor localization in the complex anatomical structure of the head and neck. One of the key advantages of these NPs is their ability to provide clearer and more detailed images [5]. Despite their promising capabilities, the risks and benefits of these NPs in MRI diagnostics must be carefully evaluated. Each type of NP exhibits unique properties that make them suitable for specific applications. Iron oxide NPs (FeNPs), for instance, are known for their strong t_2_-negative MRI contrasts, which help distinguish tumor tissues from healthy ones and make their high specificity particularly effective in imaging soft tissue tumors in the head and neck region [5]. MnNPs are notable for their dual function, providing both treatment and imaging capabilities [5].

The biocompatibility and toxicity profiles of these NPs vary [5]. FeNPs are generally considered biocompatible and are metabolized naturally in the body [5]. In contrast, Gd-based and MnNPs require careful monitoring due to potential nephrotoxicity and neurotoxicity [7]. The development of ^19^F and CuNPs are a more recent advancement, and their long-term effects in the human body are still under investigation. Based on the preclinical findings thus far, this ongoing research on novel NPs in MRI diagnostics has the potential to revolutionize cancer care, leading to more effective, personalized treatments.

In addition to novel contrast agent cores, peptide and polypeptide-based imaging platforms, with their ability to be functionalized with highly accurate targeting and responsive peptides, offer an even deeper degree of tissue visualization. These peptides can target cancer-related receptors, enzymes, and microenvironments, enhancing the precision and effectiveness of imaging, which grants these types of agents a more promising appeal for clinical translation over traditional nanoparticles [3]. For instance, targeting peptides can improve the specificity of diagnostic agents, enabling them to distinguish malignant tissues from healthy ones more effectively [3]. This targeted approach is crucial in head and neck cancer, where early and accurate diagnosis significantly impacts treatment outcomes. Responsive peptides, on the other hand, react to specific stimuli in the cancer microenvironment, such as pH changes or enzyme activities, providing real-time insights into the tumor’s behavior and response to therapy [8]. The integration of these peptide and polypeptide materials into multimodal imaging platforms also presents an opportunity to combine anatomical and functional imaging. This is particularly relevant for head and neck cancer, where detailed anatomical imaging is essential for surgical planning and assessing tumor margins, while functional imaging can provide critical information about tumor biology and treatment response.

Furthermore, recent developments in antibody and protein-based contrast agents have shown promising results in targeting head and neck tumors. Monoclonal antibodies (mAbs) and antibody fragments have been engineered to target specific tumor antigens, providing high specificity and improved imaging contrast [9]. These agents can be conjugated with imaging probes such as radionuclides or fluorophores, enabling precise localization of tumors [9]. For example, antibodies targeting the epidermal growth factor receptor (EGFR) have been developed for PET and SPECT imaging, showing potential in clinical trials [10]. Protein-based agents, including affibodies and nanobodies, offer advantages in terms of smaller size, better tissue penetration, and faster clearance from the body, making them suitable for diagnostic imaging in head and neck cancer [11].

The selected diagnostic contrast agents found within this review article include five nanoparticles, two peptides, and a small molecule. These agents can be found summarized within the table below (Table 1).

## 2. MT218 [ZD2-N3-Gd(HP-DO3A)] for the Purpose of MRI [12]

### 2.1. Background

Fibronectin (FN) is a high-molecular-weight glycoprotein in the extracellular matrix associated with cancer. Abnormal FN expression is a hallmark of cancer progression, particularly during epithelial to mesenchymal transition. However, FN is not a suitable target for molecular imaging. Instead, small peptides that target fibrin–fibronectin clots in angiogenic tumor regions have shown efficacy in animal models [13]. Notably, specific FN isoforms, like EDB-FN (extradomain B fibronectin), which is an oncofetal FN, are overexpressed in invasive cancer cells and associated with tumor invasiveness, drug resistance, and poor prognosis. EDB-FN meets the criteria for a suitable biomolecular target for developing clinically translatable targeted contrast agents (CAs) in cancer imaging, as it is highly expressed in aggressive tumors, accessible, and not abundant in normal tissues [14]. CAs targeting EDB-FN have potential applications in magnetic resonance molecular imaging (MRMI) for various cancer types, making it a promising oncotarget for clinical translation.

The design of targeted gadolinium-based contrast agents (GBCAs) for MRMI involves crucial considerations, including the selection of appropriate targeting agents. The success of clinical MRMI relies on the specific binding of these agents to molecular targets and their efficient clearance post-diagnostic imaging. Various targeting agents such as monoclonal antibodies, antibody fragments, nanobodies, and peptides, have been explored to target specific molecules like EDA-FN, EDB-FN, and fibrin-FN clots in molecular imaging and targeted drug delivery. For EDB-FN, several antibodies, fragments, and peptides have been developed. Phage display was used to identify small peptides of 7 amino acids that target the EDB protein fragment. ZD2, due to its hydrophilicity and negative overall charge, was selected as the lead peptide. Both cyclic and linear ZD2 peptides exhibited high binding affinity for EDB-FN. Binding specificity was confirmed through in vitro experiments with human cancer specimens and in vivo tests in mouse tumor models. The high specificity of ZD2 peptides was evident in various experiments, and it showed great promise for targeting EDB-FN in clinical MRMI.

Overall, this review article would like to highlight the importance of selecting the right targeting agents with the appropriate size, binding affinity, hydrophilicity, and charge for developing effective GBCAs in cancer imaging and therapy. Clinical MRI CAs fall into two main classes: linear Gd(III)-DTPA-based agents and macrocyclic Gd(III)-DOTA-based agents. Both exhibit high thermodynamic chelation stability and are efficient at altering water proton relaxation rates, but linear GBCAs have lower kinetic stability and are susceptible to releasing free Gd(III) ions by transmetalation with endogenous Zn(II), which is not ideal. Conversely, macrocyclic GBCAs are more kinetically stable and exhibit less release of free Gd(III) under normal biological conditions, making them a better choice for targeted GBCAs. 

Small peptide-targeted GBCAs have been developed to enhance cancer MRMI, with a focus on targeting fibrin-FN clots in tumors. CLT1-(Gd-DTPA) was one of the pioneering agents that demonstrated effectiveness for cancer MRMI. Other agents followed, incorporating multiple macrocyclic chelates per peptide molecule to improve targeting efficiency, such as CREKA-dL-(Gd-DOTA)4 and CLT1-G2-(Gd-DOTA), which exhibited higher r_1_ relaxivities compared to CLT1-(Gd-DTPA). Macrocyclic chelates, Gd-DOTA and Gd(HP-DO3A), along with hydroxylated Gd3N@C80, have been used to design small peptide-targeted CAs for MRMI of EDB-FN. Structures of CREKA-dL-(Gd-DOTA), CREKA-Tris(Gd-DOTA), and CLT1-G2-(Gd-DOTA) along with ZD2-(Gd-DOTA), ZD2-Gd(HP-DO3A), ZD2-Gd3N@C80, and ZD2-N3-Gd(HP-DO3A) (MT218) are shown in Figure 1. These agents, including ZD2-Gd(HP-DO3A), have shown promising results in cancer MRMI. However, clinical applications of some agents, like Gd3N@C80, face cost and production challenges, while others, like ZD2-Gd(HP-DO3A), have been optimized and synthesized at a larger scale for preclinical usage.

### 2.2. Characteristics and Clinical Applications

Relaxivities, which measure changes in water proton relaxation rates, are essential for quantifying MRI signal enhancement. Key peptide targeted GBCAs exhibit varying relaxivities. Monochelate peptide conjugates like CLT1-(Gd-DTPA), ZD2-Gd(HP-DO3A), and ZD2-(Gd-DOTA) show slightly higher r_1_ relaxivity compared to their base GBCAs. Removing the flexible spacer in ZD2-N3-Gd(HP-DO3A) increases its r_1_ relaxivity due to increased rigidity, reducing rotational rates. Tripod conjugate CREKA-Tris(Gd-DOTA)_3_ falls in between with an r_1_ relaxivity of 7.34 mM^−1^ s^−1^ per Gd.^1^ The structures of peptide dendrimeric Gd-DOTA conjugates with a molar ratio of ≥4 have r_1_ relaxivities exceeding 10 mM^−1^ s^−1^ per Gd due to their larger size, resulting in slower rotational rates. ZD2-Gd(HP-DO3A) and ZD2-N3-Gd(HP-DO3A) demonstrate modest binding affinities to the EDB protein fragment, advantageous for rapid clearance. Their kinetic stability against transmetalation, as seen with MT218, is excellent, further enhancing their potential as targeted CAs for clinical translation. 

This technology is of high spatial resolution and could be valuable for diagnostic imaging, risk stratification, active surveillance, and noninvasive evaluation of therapeutic outcomes in a range of aggressive cancers. Head and neck cancer, among other cancer types, could greatly benefit from MRMI with MT218. The overexpression of EDB-FN is associated with cancer aggressiveness and drug resistance, making MRMI a potential tool to assess the development of resistant and aggressive tumor phenotypes during treatment. This method has been effectively demonstrated, by the same researchers, in mouse models for different cancer types and offers a noninvasive means to monitor treatment responses and tumor dynamics.

### 2.3. Pharmaceutical and Safety Assessments

To meet efficacy and safety requirements, it is essential for these agents to be small in size to enable rapid penetration from the bloodstream to the target site and quick renal filtration and elimination. Additionally, factors like hydrophilicity and overall charge play a pivotal role as lipophilic and cationic agents tend to have slow clearance. Safety assessment is crucial, with a primary concern being nephrogenic systemic fibrosis (NSF) associated with GBCAs [15]. To address these concerns, pharmacokinetics, tissue clearance, and other safety aspects of MT218 needed to be investigated. MT218 exhibited a similar pharmacokinetic profile to clinical macrocyclic GBCAs, with a small molecular size. Importantly, it had the lowest brain concentration among all organs, and brain deposition was undetectable after 6 h, addressing potential safety concerns related to brain deposition of GBCAs. Repeated dosing of MT218, even at higher-than-clinical doses, showed minimal tissue accumulation in organs other than the kidneys and no detectable brain retention. Additionally, the peptide component of MT218 demonstrated stability in various biological environments. The safety profile of MT218 was further confirmed in various safety and toxicology assessments, showing no adverse effects on blood cells, immunogenicity, genotoxicity, or anaphylactic reactions. In animal studies, MT218 exhibited no significant adverse effects on cardiovascular, respiratory, or central nervous system function, with no mortality or morbidity noted. These findings support the safety and pharmacological activity of MT218, marking a significant step toward potential clinical use as an investigational new drug.

## 3. AZA-TA-Mn for the Purpose of MRI [16]

### 3.1. Background

In solid tumors, hypoxia, characterized by limited oxygen concentration and chaotic microvessel distribution, is a critical feature of the tumor microenvironment. Carbonic anhydrase, specifically CAIX, is often overexpressed as a response to hypoxia, playing a role in maintaining the pH balance within cancer cells. This makes CAIX a promising target for anticancer agents. Various inhibitors, such as acetazolamide (AZA), have been developed, and CAIX has also been explored as a biomarker for hypoxia, leading to the creation of CAIX-targeted molecular imaging probes. Given the safety concerns of traditional gadolinium-based CAs, a push for alternatives such as Mn complexes has been necessitated. These Mn complexes, particularly MnTyEDTA, have shown promise in MRI due to their safety profile and imaging capabilities. The study introduced a novel CAIX-targeted MRI probe named AZA-TA-Mn, developed by conjugating AZA and TyEDTA to a rigid triazine scaffold. The probe’s design considerations included surface binding to CAIX, higher relaxivity due to its structure, and ease of synthesis. The research aimed to investigate AZA-TA-Mn’s potential for detecting hypoxic tumors through MRI in a xenograft mouse model of ECA109 esophageal squamous cell carcinoma cells. This included a comparison with a non-specific gadolinium CA. The study also delved into the in vivo tumor selectivity of the Mn probe through inhibition experiments, tissue analysis, and immunofluorescence staining for CAIX.

In addition to AZA-TA-Mn, other ligands targeting carbonic anhydrase IX (CAIX) have been developed for imaging hypoxic tumors. These include sulfonamide-based inhibitors and, more recently, nanobodies. These ligands have been conjugated with various imaging probes for MRI, PET, and SPECT applications. For instance, sulfonamide-based ligands have shown high affinity and specificity for CAIX, enabling clear visualization of hypoxic tumor regions [17]. Nanobodies such as G250/cG250, with their small size and high binding affinity, offer rapid tumor penetration and clearance, making them ideal for real-time imaging of tumor hypoxia [17].

### 3.2. Production

The synthesis of AZA-TA-Mn, a CAIX-targeted MRI probe, involved a series of sequential steps. In the first step, Compound **1** was synthesized by reacting acetazolamide with methanol and hydrochloric acid, followed by purification. Compound **2** was then prepared by reacting N3-PEG3COOH with isobutyl chloroformate and triethylamine in anhydrous tetrahydrofuran (THF), with the addition of Compound **1** and a catalytic amount of 4-dimethylaminopyridine. Subsequently, Compound **3** was synthesized by reacting cyanouryl chloride and DIPEA in anhydrous THF and adding proparynylamine dropwise, followed by purification via silica gel column chromatography.

Compound **4** was obtained by reacting Compound **3** with TyEDTA-tert-butyl ester and cesium carbonate in anhydrous THF under reflux conditions. Compound **5** was synthesized by reacting Compound **2** with Compound **4**, TBTA, and sodium ascorbate in methanol. This step involved the deprotection of tert-butyl groups. Compound **6** was then prepared by deprotecting Compound **5** using trifluoroacetic acid. Finally, AZA-TA-Mn, the desired CAIX-targeted MRI probe, was synthesized by dissolving Compound **6** in water, adjusting the pH, and adding MnCl_2_·4H_2_O. The resulting AZA-TA-Mn was collected after lyophilization. Figure 2 shows the molecular structure of the targeted MRI probe AZA-TA-Mn.

### 3.3. Characteristics and Clinical Applications

AZA-TA-Mn exhibited a significantly higher per-Mn t_1_ r_1_ of 8.05 mM^−1^ s^−1^, outperforming the Mn-monomer and Gd-DTPA, thus enabling low-dose imaging without compromising quality. The superior r_1_ relaxivity was attributed to the slow rotation of AZA-TA-Mn due to its increased molecular weight, resulting from the rigid triazine scaffold condensed with two Mn-TyEDTA moieties and one AZA group. This enhanced r_1_ relaxivity is crucial for achieving high-quality imaging at reduced CA doses and lower systemic toxicity. Furthermore, histological analyses of ECA-109 tumor tissues confirmed a strong correlation between the selective accumulation of AZA-TA-Mn and the overexpression of CAIX on the tumor cell membrane. These findings highlight the potential of AZA-TA-Mn for targeted imaging in the presence of negatively charged tumor cell membranes and suggest that the overexpression of CAIX on tumor cell surfaces can facilitate the binding between CAIX and negatively charged Mn probes.

Initially, both the Gd-DTPA control and AZA-TA-Mn showed moderate tumor enhancement, with no significant difference in contrast-to-noise ratio (CNR) during the first 8 min after administration. However, AZA-TA-Mn displayed a prolonged and persistent enhancement pattern, with a significantly higher CNR at 30 min compared to Gd-DTPA. Importantly, AZA-TA-Mn exhibited moderate enhancement within the tumor core after 30 min, which was absent in the Gd-DTPA images. The area-under-the-curve analysis between 30 and 60 min confirmed AZA-TA-Mn’s superior performance. The prolonged and persistent enhancement, observed at half the dose of Gd-DTPA but with approximately 1.58 times the peak CNR, suggested that the incorporation of the CAIX-targeting AZA motif into the triazine scaffold enhanced the accumulation of the Mn probe in tumor tissues. Non-specific enhancement was noted, likely due to AZA-TA-Mn accumulation in clearance organs near the tumor. To validate specific binding to CAIX, a competition study was conducted, co-administering excess free AZA along with AZA-TA-Mn. This resulted in a pronounced reduction in tumor contrast enhancement, confirming the specific binding of AZA-TA-Mn to esophageal squamous cell carcinoma (ESCC) tumors. These findings indicate that AZA-TA-Mn holds promise as a specific and effective MRI CA for detecting hypoxic tumor regions in ESCC.

### 3.4. Pharmaceutical and Safety Assessments

In a competitive binding experiment, the tumor accumulation of the Mn probe, AZA-TA-Mn, was quantitatively assessed using inductively coupled plasma mass spectrometry. After injecting AZA-TA-Mn into tumor-bearing mice, tissues from the tumor and other organs were collected at 60 min post-injection and analyzed. The results indicated that AZA-TA-Mn could be eliminated through mixed hepatobiliary and renal pathways. Importantly, co-administering excess free AZA significantly reduced the accumulation of AZA-TA-Mn in tumor tissues. The decreased Mn levels in the tumor tissues of the AZA(+) group confirmed the selective targeting of tumor tissues by AZA-TA-Mn, consistent with the observation of decreased contrast enhancement in the competitive binding imaging study. This provides strong evidence of AZA-TA-Mn’s ability to selectively accumulate in tumor tissues. While the bioaccumulation profile on this CA is preferable, more investigation must be done to assess its in vivo safety profile and clearance pathways.

## 4. KMnF3/Yb/Er (Biotin/PEG-UCNPs) for the Purpose of MRI [18]

### 4.1. Background and Production

Glioblastoma (GBM) is a highly aggressive and invasive brain tumor associated with high morbidity and mortality. Accurate delineation of GBM invasion is crucial for effective surgical resection and increased patient survival. While MRI is the preferred modality for preoperative assessment, the performance of clinically available MR GBCAs is suboptimal, especially for imaging the infiltrating edges of GBM. Furthermore, concerns have arisen about the safety of GBCAs, referring to their association with NSF. To address these limitations, Mn-based NPs have gained attention as alternative CAs due to their improved imaging performance. In particular, MnNPs with enhanced r_1_ relaxivity are desirable for precise GBM margin delineation. In this context, biotin, a growth micronutrient, serves as a promising targeting moiety by exploiting the sodium-dependent multivitamin transporter (SMVT), overexpressed on the surfaces of many cancer cells [19]. This study focuses on the development of upconversion nanoparticles (UCNPs) featuring high r_1_ relaxivity and single-band upconversion. Biotin-conjugated PEGylated UCNPs (biotin/PEG-UCNPs) were specifically designed to target GBM, potentially revolutionizing t_1_ MRI for accurate delineation of invasive tumor margins. The research demonstrated the low toxicity and excellent biocompatibility of biotin/PEG-UCNPs, highlighting their potential for MRI and guiding future excision procedures.

These UCNPs utilize upconversion luminescence (UCL), which refers to a process where low-energy photons, such as infrared light, are converted into higher-energy photons, typically in the visible or ultraviolet spectrum, thus permitting efficient and precise detection of tumors. Accurate tumor margin detection is crucial in treating aggressive brain tumors such as GBM. This precision is essential for surgical resection to remove cancerous tissue while sparing healthy brain regions, preventing recurrence. It is equally vital in radiation therapy to avoid harm to normal tissue and optimize chemotherapy by facilitating targeted drug delivery. Additionally, monitoring treatment response relies on assessing changes in tumor margins, influencing treatment decisions and prognosis assessment. Innovative imaging techniques, like biotin/PEG-UCNPs as MR contrast agents, show promise in improving margin detection, with potential significant impacts on patient outcomes.

The synthesis of UCNPs capped with biotin (biotin/PEG-UCNPs) began with the development of oleic acid (OA)-stabilized UCNPs (KMnF_3_:18% Yb, 2% Er) using a hydrothermal method. Subsequently, the hydrophobic OA phase of UCNPs was converted into a water-soluble phase via hydrochloric acid treatment, eliminating OA and forming OA-free UCNPs. These were then decorated with NH_2_-PEG2K-SH through intense thiol-metal attraction to create PEG-UCNPs. Finally, biotin was added to cap the PEG-UCNPs. Successful conjugation was confirmed by the disappearance of OA characteristic bands and the emergence of PEG bands in Fourier-transform infrared (FTIR) spectra.

### 4.2. Characteristics and Clinical Applications

The morphological examination using transmission electron microscopy (TEM) revealed that OA-UCNPs and biotin/PEG-UCNPs both had uniform, cube-like structures, and their morphology remained consistent with bare UCNPs.

High-resolution TEM images indicated that the lattice fringes were at a distance of 0.614 nm, corresponding to the (100) lattice plane of the rectangular phase. Energy-dispersive X-ray spectrometry confirmed the presence of K, Mn, Yb, and F elements, supporting the elemental composition. Dynamic light scattering (DLS) measurements indicated the hydrodynamic diameters of OA-free UCNPs, PEG-UCNPs, and biotin/PEG-UCNPs were 145.3, 186.9, and 197.8 nm, respectively. The larger size of hydrophilic UCNPs was attributed to surface modification and the relatively large NP size, facilitating their retention within the tumor region. The zeta potential of UCNPs was evaluated, with OA-UCNPs having a positive potential of 26.2 mV, PEG-UCNPs showing a negative potential of −7.8 mV due to the PEG layer, and biotin/PEG-UCNPs possessing a −13.8 mV zeta potential after biotin conjugation. The results indicated the successful surface modification and conjugation of biotin and PEG molecules on the UCNPs. Further, the study evaluated the stability of biotin/PEG-UCNPs over seven days, demonstrating good water solubility and stability. X-ray powder diffraction patterns confirmed the crystallinity of the synthesized NPs, and FTIR verified the exchange of OA by PEG and the covalent conjugation of biotin.

Confocal laser scanning microscopy (CLSM) revealed enhanced cellular uptake of biotin/PEG-UCNPs by GL261 cells. These NPs were observed as strong red fluorescence surrounding the cell nuclei, signifying their uptake into the cytoplasm without entering the nucleus. In contrast, passive uptake was noted with PEG-UCNPs. A blocking assay with high-dose biotin further demonstrated the specific binding of biotin/PEG-UCNPs to GL261 cells via the biotin receptor. Cytotoxicity assessments showed that both PEG-UCNPs and biotin/PEG-UCNPs had minimal impact on cell survival and viability, highlighting their low toxicity and excellent biocompatibility. Live cell staining confirmed the well-preserved viability of most cells treated with biotin/PEG-UCNPs.

Biotin/PEG-UCNPs exhibited a high r_1_ relaxivity of 6.124 mM^−1^ s, and their longitudinal relativity time correlated with the proximity of Mn ions to neighboring water protons, demonstrating their potential as efficient t_1_ CA. These NPs demonstrated a concentration-dependent brightening effect without affecting the r_1_ value significantly. Additionally, the r_1_ value of biotin/PEG-UCNPs was approximately 2.04 times higher than clinically used GBCAs. MRI imaging further highlighted the potential of biotin/PEG-UCNPs as effective t_1_ CAs, indicating their utility for improved imaging with relatively lower quantities and reduced side effects.

To assess transport characteristics and efficacy of biotin as a tumor target, in vitro, in vivo, and ex vivo studies were performed. In vitro experiments involved the creation of a blood–brain barrier (BBB) model using bEnd.3 cells. The findings demonstrated that biotin/PEG-UCNPs displayed notably superior transport efficiency when compared to non-targeting PEG-UCNPs. This underscores the critical role played by biotin in facilitating BBB-crossing transcytosis, mainly through the SMVT. Moving to in vivo assessments, MRI was performed in a mouse model bearing GBM. Biotin/PEG-UCNPs led to a substantial improvement in t_1_-weighted MRI contrast, thereby enhancing the delineation of the tumor’s invasive boundary compared to the non-targeting NP group. The relative signal intensity (rSI) in the targeting group can be up to 2.39 times higher at 1 h compared to the non-targeting group, demonstrating the strong binding affinity of biotin towards GL261 glioblastoma cells (*n* = 3). The ex vivo imaging further substantiated the distribution of biotin/PEG-UCNPs within the infiltrating tumor margin, aligning with the histological findings. 

While in vivo upconversion luminescence (UCL) signals were somewhat limited, ex vivo UCL signals were detectable in the biotin/PEG-UCNPs group. Collectively, these findings suggest that biotin/PEG-UCNPs hold promise as dual-mode imaging probes, enabling the identification of the invasive margin of glioblastoma.

### 4.3. Pharmaceutical and Safety Assessments

To evaluate the clinical potential of biotin/PEG-UCNPs as MR CAs, their metabolic pattern was studied in Kunming mice. Whole-body MR imaging showed increased kidney illumination at 4 h post-injection compared to the pre-injection state, with the signal nearly disappearing after 24 h. This pattern suggests that the primary route of biotin/PEG-UCNP elimination is through the kidneys, indicating their potential as safe and metabolizable clinical CAs. This is a crucial prerequisite for clinical applications, as it demonstrates their ability to be completely metabolized in the body without causing evident toxicity over time.

Additionally, histopathological examinations, hematological tests, and body weight measurements were conducted after intravenous injection of 5 mg/kg biotin/PEG-UCNPs at acute, subacute, and chronic stages (3, 15, and 30 days). The results showed no adverse effects on liver, kidney, blood parameters, or body weight. There was also no observable tissue damage in vital organs, including the brain, indicating good biocompatibility of biotin/PEG-UCNPs [18]. These findings indicate that biotin/PEG-UCNPs are safe for in vivo use as medical imaging CAs; however, long-term assessments are still needed before clinical application.

## 5. Gd-SA for the Purpose of MRI [20]

### 5.1. Background and Production

Single-atom catalysts have gained attention for their enzymatic properties, stability, and biocompatibility, but most Gd atoms within these NPs are inaccessible to water molecules, reducing their efficiency [21]. The single-atom Gd nano-contrast agents (Gd-SAs) developed in this study exhibited exceptional catalytic properties and can be anchored onto porous nanomaterials like hollow N-doped carbon nanospheres, which can potentially address the limited accessibility of water to Gd atoms. The researchers focused on designing a GD-SA for enhanced MRI to create a novel, efficient, and biocompatible MRI CA, potentially revolutionizing disease diagnosis with MRI. 

Gd-SA was created by coordinating Gd atoms with six N atoms and two O atoms on hollow N-doped carbon nanospheres. The exact synthesis of Gd-SA involved several steps, with a basic outline shown in Figure 3. Initially, 90 nm SiO_2_ nanospheres were synthesized. These nanospheres served as templates for Gd-SA fabrication. Dopamine and Gd^3+^ ions were added to the SiO_2_ nanospheres, creating a core–shell structure called SiO_2_@Gd/PDA. Subsequently, the polydopamine shell was converted into N-doped carbon at 900 °C under an Ar atmosphere, while Gd atoms were anchored by N and O atoms in a single-atom configuration. The inner SiO_2_ core was removed using hot sodium hydroxide solution, and Gd-SA was washed thoroughly with diluted hydrochloric acid and deionized water to eliminate residual sodium hydroxide. Finally, DSPE-PEG_2000_-NH_2_ was added to enhance solubility for further biological assessments. 

### 5.2. Characteristics

The synthesized Gd-SA exhibited uniform hollow spherical morphology with an average outer diameter of 107 ± 9 nm, as observed in scanning electron microscopy and TEM images. High-resolution TEM revealed a hollow sphere with a thickness of 6.5 ± 0.5 nm and confirmed the absence of Gd clusters. Energy-dispersive spectroscopy mapping showed homogeneous distributions of O, N, and Gd species on the hollow carbon spheres. The N and O atoms in Gd-SA originated from the dopamine monomer’s hydroxy and amino groups, respectively. Aberration-corrected high-angle annular dark-field scanning TEM images directly revealed isolated single Gd atoms.

The content of Gd atoms in Gd-SA was determined to be 0.42 wt% by inductively coupled plasma optical emission spectrometry. Structural characterization via powder X-ray diffraction (XRD) analysis indicated that Gd-SA’s pattern was similar to pure N-doped carbon, distinct from Gd_2_O_3_ NPs. Raman spectroscopy showed minor differences in the intensity ratio of the D-band to the G-band (ID/IG) between Gd-SA and the N-doped carbon support, indicating that Gd loading did not alter the basic structure of the support. Gd-SA possessed a large specific surface area (1112.4 m^2^/g) and an average pore diameter of 6.4 nm, as determined by N_2_ adsorption–desorption isotherms and BET analysis. FTIR indicated the presence of hydroxy groups on Gd-SA’s surface and confirmed the successful coating of DSPE-PEG_2000_-NH2.

X-ray photoelectron spectroscopy (XPS) analysis showed the presence of C, N, O, and Gd elements in Gd-SA. The Gd 4d XPS spectrum exhibited characteristic peaks at binding energies of 147.6 and 142.8 eV, corresponding to Gd 4d_2/3_ and Gd 4d_5/2_, respectively. High-resolution N 1s XPS revealed four nitrogen species. The content of pyrrolic and pyridinic nitrogen species was as high as 36.0%, contributing to the stabilization of single Gd atoms. The C 1s XPS spectrum indicated the presence of C-N bonds. High-resolution O 1s XPS showed peaks related to O vacancies, C-OH, and Gd-OH groups.

Synchrotron radiation-based Gd L_3_-edge X-ray absorption fine structure spectroscopy confirmed the existence of single Gd atoms in Gd-SA. Wavelet transform analysis further supported the absence of metallic Gd or GdO_x_ species. Density functional theory (DFT) calculations revealed that the adsorption energy of a water molecule on Gd-SA was lower compared to Gd-DTPA, indicating weaker bonding and faster water exchange on Gd-SA. The charge density difference analysis demonstrated charge transfer between the O atom and the Gd atom, with charge accumulation mainly around Gd-SA’s Gd and O atoms, supporting the absence of hydrogen bonding in Gd-SA compared to Gd-DTPA. This lack of hydrogen bonding in Gd-SA’s core likely contributes to its favorable MRI properties due to having a faster water exchange rate.

Additionally, Gd-SA exhibited superior relaxivity properties in comparison to Gd-DTPA. At a high magnetic field of 7 Tesla, Gd-SA displayed significantly brighter t_1_-weighted MRI, even with lower Gd doses. Its r_1_ was remarkably high at 11.05 mM^−1^ s^−1^, approximately 3.6 times greater than that of Gd-DTPA (3.03 mM^−1^ s^−1^). This high r_1_ value, combined with a favorable r_2_/r_1_ ratio of 9.3, suggests Gd-SA’s potential for t_1_-t_2_ dual-mode MRI imaging. The excellent relaxivity was attributed to the effective water molecule exchange rate facilitated by the weaker Gd-water bonding in Gd-SA, which resulted from its single-atom Gd coordination. These relaxivity findings highlight Gd-SA as a promising MRI CA with enhanced imaging capabilities.

### 5.3. Clinical Applications

Gd-SA demonstrated remarkable MRI tumor imaging capabilities. In vivo studies showed that Gd-SA significantly enhanced the visibility of tumors compared to Gd-DTPA, achieving a maximum tumor-to-normal tissue contrast ratio of 1.61 at 2 h post-injection. What is particularly noteworthy is Gd-SA’s ability to provide clear and distinct boundaries of the tumor, improving diagnostic precision. Even in the case of very small tumors, approximately 3 mm in diameter, Gd-SA exhibited superior performance with a higher tumor-to-normal tissue contrast ratio of 1.52 at just 1 h post-injection, outperforming Gd-DTPA, which reached only 1.17. This enhanced imaging capacity is crucial for early tumor detection, offering improved chances for successful treatment. Gd-SA’s exceptional tumor-imaging capabilities, including its ability to enhance contrast and delineate tumor boundaries, make it a promising candidate for high-quality and precise MRI-based tumor diagnosis and monitoring.

While these NPs have not yet been tested in the context of specific tumor targeting capabilities, they hold promise by merit of their versatile applications to become functionalized in a variety of ways [22]. One approach involves modifying the nanoparticle surfaces with ligands or antibodies that specifically recognize biomarkers or receptors overexpressed on the surface of certain brain tumor cells [22]. For example, antibodies targeting epidermal growth factor receptor variant III (EGFRvIII) or vascular endothelial growth factor receptor (VEGFR) can be conjugated to Gd-SAs to enhance their affinity for GBM cells [22]. Moreover, Gd-SAs can serve as carriers for therapeutic agents, allowing for targeted drug delivery to brain tumor sites while minimizing systemic exposure to chemotherapy agents and reducing side effects [22]. This approach is especially relevant in the context of brain tumors that are difficult to access due to the BBB. 

Functionalization with BBB-penetrating peptides or ligands can facilitate the passage of Gd-SAs into the brain, increasing their access to tumor sites [22]. In addition to their role in drug delivery and MRI contrast enhancement, Gd-SAs can be equipped with multiple imaging moieties, such as fluorescent dyes or radionuclides. This enables multimodal imaging, allowing for the precise localization of brain tumors using different imaging techniques simultaneously [22]. Responsive targeting can also be achieved by engineering Gd-SAs to respond to specific conditions within the tumor microenvironment, ensuring the release of therapeutic agents precisely where and when needed [22]. By tailoring the nanoparticles to recognize and interact with unique molecular signatures of brain tumors, Gd-SAs can improve the accuracy of diagnosis, treatment, and monitoring, ultimately leading to better outcomes for patients with brain tumors.

### 5.4. Pharmaceutical and Safety Assessments

The in vitro and in vivo safety assessments of Gd-SA were conducted to evaluate its biocompatibility and potential as an MRI CA. In vitro studies revealed excellent stability, with only 0.91 wt% of Gd leached from Gd-SA under neutral pH conditions, significantly lower than Gd-DTPA and Gd_2_O_3_. Gd-SA exhibited low cytotoxicity to human umbilical vein endothelial cells and minimal hemolysis on mouse red blood cells. In vivo biodistribution studies showed that Gd-SA primarily accumulated in the liver, whereas Gd-DTPA accumulated mainly in the kidneys, indicating different clearance pathways. Gd-SA demonstrated enhanced tumor accumulation compared to Gd-DTPA, making it promising for tumor MRI. Importantly, Gd-SA reduced kidney accumulation, suggesting a lower risk of kidney dysfunction. In vivo biosafety assessments, including blood biochemistry and histological examination, demonstrated negligible harm to hematology, liver function, kidney function, and major organs. These findings collectively highlight the biocompatibility and excellent tumor-imaging capabilities of Gd-SA, positioning it as a promising MRI CA for potential clinical applications.

## 6. IN-ABPs for the Purpose of CT [23]

### 6.1. Background

Research on the development of targeted CA for CT imaging aims to enhance the specificity and sensitivity of CT imaging in medical diagnostics. Conventional iodinated CT CAs, often used in clinical practice, have limitations. These can include rapid clearance from the body and non-specific accumulation in organs or tissues [24]. Other NPs offer promising alternatives, with advantages such as specific molecular targeting capabilities, prolonged circulation times in the blood, and controlled clearance pathways [25]. NPs with high electron density, like those containing gold or bismuth, show potential as CT CAs due to their low rates of renal clearance and extended residence in the vascular system [26,27,28]. Additionally, NPs and macromolecular agents within the nanoscale range tend to accumulate in solid tumors, capitalizing on the enhanced permeability and retention effect [29].

Polymeric dendrimers with iodine cores are highlighted here as structures with chemical functionalities that can efficiently carry therapeutic agents and contrast materials. Dendrimers as carriers for CT CAs could potentially allow for more extended blood pool imaging. Research groups have explored the development of iodine-based CT probes to improve early disease detection, and these probes can be targeted to highly expressed biological markers, resulting in densely loaded CAs for CT molecular imaging. While multiple approaches have been optimized for iodinated NPs as CT contrast media, none of these compounds have yet been clinically approved, despite the potential for better CT imaging in the future. Past research on this underscores the importance of developing highly iodinated compounds for improving CT contrast, potentially opening new avenues for early disease diagnosis in clinical practice.

The researchers highlighted here recently showcased an innovative strategy to develop iodine-based CT CAs for cancer detection, utilizing activity-based probes (ABPs). Rather than targeting proteins directly, the focus is on specific enzymatic activity, particularly cysteine cathepsin proteases that are highly active in diseases with a high macrophage content, like cancer and atherosclerosis [30]. This customization allows for precise targeting, ensuring that the contrast agent predominantly accumulates in tissues such as those found in cancers of the head and neck. ABPs consist of a recognition element that imparts selectivity to the protease, a CA, and a “warhead” for covalently linking the probe to its target. The researchers created iodinated nanoscale activity-based probes (IN-ABPs) targeted to these cathepsin proteases, including cathepsin B and L. They incorporated iodine tags onto these probes to enhance their CT contrast properties. The basic structure of G1 IN-ABPs and PAMAM G3 IN-ABPs is shown in Figure 4.

### 6.2. Production

In the design and synthesis of ABPs, the researchers developed molecules with a recognition element tailored for cathepsin proteases and a covalent “warhead”. GB111-NH2 served as the ABP core, known for its high selectivity towards cathepsin B, L, and S. To introduce iodine tags, they generated these tags using commercially available compounds, namely, 2,3,5-triiodobenzoic acid (TBA) and iopanoic acid (IPA). TBA was transformed into a succinimidyl ester (SE), while IPA’s free amine was initially acetylated and then converted into SE. These activated iodine tags were subsequently linked to the ABP core. For the creation of single and multiple tagged IN-ABPs, single-iodine-tagged ABPs were formed by directly attaching the iodine tags to GB111-NH2. In contrast, multiple-tagged IN-ABPs were produced by reacting the iodine tags with a GB111-NH2 derivative extended with glutaric acid, enabling the attachment of several iodine tags. The iodine tags were connected through amide bonds. In the synthesis of PAMAM dendrimers, PAMAM dendrimers served as nanocarriers for the IN-ABPs, particularly PAMAM G1 and G3 core compounds. These dendrimers were prepared by reacting PAMAM cores with the SE of the IPA tag to yield iodine-tagged dendrimers. GB111-NH2 was then linked to form ABPs for these dendrimers. And lastly, some PAMAM G3 compounds had their remaining free amines modified with acetyl or PEG groups, while others were left unaltered. Cy5 labels were incorporated into these compounds, either directly or after acetylation or PEGylation.

### 6.3. Characteristics and Clinical Applications

The researchers developed a library of IN-ABPs, tailoring them to target specific cathepsin proteases (B or L), which were also designed with iodine tags to enhance their CT contrast properties. The iodine tags were linked to the ABP core and tested for their ability to bind to recombinant human cathepsins and inhibit their activity. The study also evaluated the probes’ ability to permeate cells and label cellular cathepsins. In competitive inhibition assays performed on intact cells, the IN-ABPs effectively inhibited cellular cathepsins. Cell permeability and the probes’ ability to label cellular cathepsins were evaluated using competitive inhibition assays on NIH-3T3 cells. The results showed that probes with IPA iodine tags exhibited better inhibition and cell permeability, leading to the decision to use IPA tags for further probe development. More advanced IN-ABPs were developed, featuring up to 48 iodine atoms (16 tags) based on PAMAM (polyamidoamine) G1 and G3 cores. For PAMAM G1 compounds, the dendrimer core was tagged with IPA in basic conditions, yielding multiple iodine-tagged dendrimers. These dendrimers were further functionalized, including the addition of a Cy5 label, creating a range of CAs for cellular and in vivo studies. The design extended to PAMAM G3 dendrimers with solubility constraints, as the highly cationic PAMAM dendrimers could impact cellular membranes. These dendrimers featured a controlled ratio of acetyl, PEGylated, or free amines to enhance their compatibility with cellular structures. The researcher’s work here encompasses a wide array of CAs, featuring various iodine tags and dendrimer cores, ready for applications in cellular and molecular imaging, with a focus on cancer detection. 

The conventional iodinated CT CAs used in clinical practice are low-molecular-weight compounds that tend to accumulate non-specifically in various organs and tissues. Additionally, these agents are rapidly cleared from the body, limiting their imaging window to a matter of minutes or even seconds. Through in vivo experiments using tumor-bearing mice, the researchers compared the performance of the targeted IN-ABPs to their non-targeted analogs in detecting cathepsin activity within tumors. These experiments demonstrated that the covalent nature of the IN-ABPs allowed for extended probe retention in tumor tissues, leading to detectable CT signals. Overall, the results indicated that the targeted IN-ABPs (HG92 and HG90) outperformed the non-targeted controls (HG31 and HG99, respectively) in terms of their ability to specifically detect cathepsin activity within tumor tissues. The targeted CAs demonstrated the potential for highly sensitive and specific cancer detection using CT imaging, offering promising prospects for early disease diagnosis. 

### 6.4. Pharmaceutical and Safety Assessments

Before proceeding to in vivo experiments, the researchers conducted a cytotoxicity assessment on PAMAM dendrimeric compounds using a methylene blue cell viability assay. They observed that two experimental IN-ABPs, namely, HG90 and HG92, exhibited no signs of toxicity. Therefore, these compounds were chosen for further in vivo studies. However, two compounds with multiple free amines (HG87 and HG82) and two acetylated compounds (HG99 and HG86) led to a 15–35% reduction in cell viability at the highest concentration of 10 μM after 48 h. In general, the majority of the compounds tested showed minimal toxicity within the concentration range under examination.

## 7. NaHoF_4_ for the Purpose of CT/MRI [31]

### 7.1. Background and Production

Recent advancements have seen the development of several MR/CT imaging agents, such as FeBiNPs, Gd-chelated AuNPs, and Au or TaOx-decorated Fe_3_O_4_NPs [32,33,34,35,36,37,38]. With MRI moving towards ultra-high fields like 7.0T, there is a growing need for high-performance ultra-high field MR/CT dual-modality imaging CAs. The study focuses on dysprosium (Dy^3+^) and holmium (Ho^3+^) ions as promising candidates for ultra-high-field MRI CAs due to their high effective magnetic moments. Ho^3+^ ions, in particular, exhibit higher attenuation characteristics than iodine, making them suitable for both CT and ultra-high-field MRI [39]. This study introduced NaHoF_4_ NPs of various sizes as dual-modality CAs for ultra-high-field MRI and CT. Comprehensive evaluations were conducted to assess their morphology, size, X-ray absorption, magnetic properties, biocompatibility, and in vivo ultra-high-field MR/CT imaging performance. Unlike previous studies that only presented basic relaxivity data, this research delves deeper into the relaxation mechanisms of different sizes of NaHoF_4_NPs, combining experimental results with theoretical analysis. These findings aim to advance the medical translation of these nano-CAs for ultra-high field MR/CT dual-modality imaging applications.

In this study, uniform NaHoF_4_ NPs of various sizes were synthesized using a thermal decomposition method, with controlled nucleation and growth achieved by adjusting reaction time and temperature. The synthesis of OA-NaHoF_4_ NPs involved dissolving 2 mmol of HoCl_3_·6H_2_O in a mixture of deionized water, oleic acid, and 1-octadecene. This mixture was stirred at room temperature, then heated to remove water and maintained at a higher temperature to achieve the desired reaction. After cooling, a methanol solution with NaOH and NH_4_F was added and stirred further. Post-methanol evaporation, the solution was heated again and then cooled down. The resulting product was washed with ethanol and dispersed in chloroform. Variations in ligand concentration, temperature, and time are made to synthesize NaHoF_4_ NPs of different sizes (~8, ~13, and ~29 nm). For biocompatibility, the NaHoF_4_ NPs underwent surface modification with DSPE-PEG_5k_. This involved mixing NaHoF_4_ solution with DSPE-PEG_5k_ solution, followed by incubation and evaporation of the solvent under vacuum. Finally, water was added and sonicated to obtain PEGylated NaHoF_4_ NPs.

### 7.2. Characteristics and Clinical Applications

High-resolution TEM confirmed their crystalline nature, and energy-dispersive X-ray spectroscopy identified the presence of essential chemical elements. Sodium, as part of the NaHoF_4_ formula, helps in maintaining charge balance and stabilizing the NPs. Its ionic nature is key to the structural formation of these NPs. Holmium, a rare earth element, is central to the magnetic characteristics of the NPs. Its paramagnetic properties are highly valuable, especially for MRI contrast enhancement, making these NPs effective for magnetic resonance imaging. Fluorine atoms form fluorides with holmium and sodium, contributing to the structural framework of the NPs. This element also influences the NPs’ interactions with X-ray radiation, as observed in CT imaging applications. The synergy of these elements in NaHoF_4_ NPs thus endows them with unique properties that are beneficial for dual-modality imaging techniques, particularly in MRI and CT scans.

For biomedical applications, DSPE-PEG_5000_ was used to enhance the biocompatibility of the NPs. This modification maintained their morphology and dispersity in water. DLS results indicated hydrodynamic diameters ranging from ~13 to ~56 nm, suggesting a ~5 nm thick PEG layer encapsulating the NPs. X-ray absorption and ultra-high-field MRI experiments were conducted using aqueous solutions of various Ho^3+^ ion concentrations. Phantom CT images at 120 KVp showed a sharp signal enhancement with increased Ho^3+^ concentration. The HU values for PEG-NaHoF_4_ NPs were about 42.1 HU L/g, substantially higher than the clinically used iobitridol (16.5 HU L/g) and nearly double that of WS_2_ nanosheets (22.01 HU L/g). The magnetic properties of the NPs were evaluated through zero-field-cooled/field-cooled curves and magnetization curves at both 300 K and 5 K. These studies indicated that the NPs were paramagnetic with magnetization values at 30 KOe of 3.73, 4.29, 4.61, and 9.48 emu/g for 3 nm, 7 nm, 13 nm, and 29 nm NPs, respectively.

In vitro t_2_-weighted imaging at 1.5, 3.0, and 7.0 T revealed concentration- and size-dependent MR contrast performance. The r_2_, r_1_, and r_2_/r_1_ ratios of NaHoF_4_ NPs varied with magnetic fields. Notably, the r_2_ value of 29 nm NaHoF_4_ NPs at 7.0 T was 222.64 mM^−1^s^−1^, higher than previously reported NaDyF_4_ NPs and 70 nm Dy_2_O_3_ NPs. The r_2_/r_1_ ratios were significantly higher than Feridex and Resovist, indicating the suitability of NaHoF_4_ NPs for ultra-high-field MRI.

Relaxation mechanisms of NaHoF_4_ NPs, essential for understanding their behavior in MRI, include dipolar and Curie contributions. The Curie contribution, dominant in particles smaller than 7 nm, is based on the interaction between water protons and a large static magnetic moment arising from electrons. The relaxation properties of NaHoF_4_ NPs align with the square of the increment of magnetic field strength and increase remarkably as the diffusion correlation time increases, except for the smallest 3 nm NPs.

In vivo imaging applications were explored as well, and they showed excellent CT/MR dual-mode contrast imaging. In vivo CT imaging in mice bearing 4T1 murine breast cancer tumors revealed a notable brightness in tumor regions post-injection, with HU values increasing from 36.6 ± 11.1 to 209.8 ± 40.1. Ultra-high-field MRI at 7.0 T also demonstrated high negative contrast in the kidneys over 1 h, indicating the efficacy of NaHoF4 NPs in dual-modality imaging applications.

### 7.3. Pharmaceutical and Safety Assessments

The biocompatibility of NaHoF4 NPs was also thoroughly investigated. Hemolysis experiments and MTT assays against human glioblastoma cells U87MG demonstrated low cytotoxicity. In vivo studies on mice showed no significant alterations in liver function markers (ALT, AST, AKP), kidney function indicators (CRE, BUN), or vital blood parameters (WBC, RBC, LYM, PLT, HGB, MCV) compared to control groups. No appreciable adverse effects on tissues were observed, confirming the NPs’ biocompatibility.

In summary, NaHoF4 NPs, with their unique size-dependent properties, excellent biocompatibility, and high contrast efficiency, hold significant potential as CAs for future ultra-high-field MR/CT dual-modality imaging applications. And while the contrast efficiency was evaluated in cancer of the breast, these properties could be leveraged for identifying small lesions and assessing tumor infiltration in soft tissues of the head and neck. Their potential for functionalization with targeting moieties also presents opportunities for more precise tumor localization. The safety profile of NaHoF_4_ NPs, demonstrated through in vivo studies, is particularly important given the sensitivity of the head and neck area. Importantly, their capability for simultaneous MRI and CT imaging streamlines the diagnostic process, offering comprehensive anatomical and soft tissue details in a single session, invaluable in accurately staging and planning treatment for cancers of this region. The detailed exploration of their physical and biological characteristics underscores their promise in enhancing diagnostic capabilities in medical imaging.

## 8. [^68^Ga]Ga-HX01 for the Purpose of PET/CT [40]

### 8.1. Background and Production

Dual targeting strategies, particularly focusing on tumor vasculature, offer a promising approach to overcome these limitations. Two specific targets, integrin α_v_β_3_ and aminopeptidase N (APN or CD13), are highlighted due to their overexpression on both tumor cells and neovasculature, making them ideal for developing radioligands [41,42]. Peptides, with advantages like easy synthesis, chemical modification, and low immunogenicity, are considered ideal ligands for targeted diagnosis and therapy. The RGD and NGR peptide sequences are well-known for targeting integrin α_v_β_3_ and CD13, respectively [42,43]. The article introduces the development of a heterodimeric tracer, HX01, targeting both integrin α_v_β_3_ and CD13, which has shown promise in previous studies. The goal was to evaluate HX01’s clinical applicability, its potential for radiation therapy, and its use as a theranostic pair with ^68^Ga for diagnosis and ^177^Lu for therapy. The present review article will focus mostly on this new agent’s diagnostic properties. This study aims to improve the probe’s capabilities, confirm its specificity, assess optimal imaging doses, and evaluate its diagnostic performance across various tumor models, paving the way for clinical translation. [^68^Ga]Ga-HX01 has received approval for a phase I clinical study by the National Medical Production Administration.

The radiolabeling of HX01, RAD-NGR, and RGD-NAR was conducted using previously established methods with slight modifications. [^68^Ga]GaCl_3_ was obtained from a ^68^Ge/^68^Ga radionuclide generator and eluted with 0.05 M HCl. For HX01 radiolabeling, 60 μg of lyophilized HX01 powder was redissolved in 1 mL of 0.25 M sodium acetate solution. Subsequently, 4 mL of [^68^Ga]GaCl_3_ eluent was added, and the mixture was heated at 60 °C for 10 min, maintaining a pH of 3.5–4. After cooling to room temperature, the reaction mixture was processed using a C18 light Sep-Pak cartridge, washed with saline, and the final product was obtained by elution with 45% ethanol followed by saline and filtration through a 0.22 μm pore size filter.

### 8.2. Characteristics and Clinical Applications

In this research, [^68^Ga]Ga-HX01 PET/CT imaging played a central role in evaluating its potential as an effective imaging agent for various tumor types. The study employed the InliView-3000B small animal PET/SPECT/CT system (Novel Medical, Beijing, China). To validate the specificity of [^68^Ga]Ga-HX01, a blocking study was conducted. Excessive amounts of unlabeled agents, including RGD, NGR, RGD + NGR, and HX01 (each at 12.5 mg/kg per mouse), were co-administered with [^68^Ga]Ga-HX01. This study aimed to demonstrate that the tumor uptake of [^68^Ga]Ga-HX01 could be significantly reduced when specific binding sites were blocked by these excess agents. The results from this blocking study provided clear evidence of the specificity of [^68^Ga]Ga-HX01 for its target receptors.

A single target imaging study was also performed using partly muted tracers, [^68^Ga]Ga-RGD-NAR and [^68^Ga]Ga-RADNGR, which possessed similar molecular weights to [^68^Ga]Ga-HX01. The rationale behind these studies was to confirm that [^68^Ga]Ga-HX01 exhibited increased affinity and higher tumor uptake compared to tracers with similar characteristics but targeting single receptors. The results of these single target studies supported the superior targeting capability of [^68^Ga]Ga-HX01.

The researchers also conducted dynamic [^68^Ga]Ga-HX01 PET/CT imaging to investigate its distribution and metabolic behavior in the BxPC-3 xenograft model. This dynamic imaging revealed rapid distribution, with the main route of elimination being through the kidneys. The tumor uptake of [^68^Ga]Ga-HX01 gradually increased, reaching its peak at approximately 10 min post-injection, followed by a slower rate of decline. The uptake was approximately 1.395 ± 0.131%ID/g, significantly higher than [^68^Ga]Ga-RGD-NAR (0.927 ± 0.146%ID/g) and [^68^Ga]Ga-RAD-NGR (0.244 ± 0.051%ID/g). These dynamic data offered insights into the agent’s behavior within the body.

A dose-climbing study was conducted to determine the sensitivity of [^68^Ga]Ga-HX01 for detecting tumors at different dose levels. The study demonstrated that even at low doses, [^68^Ga]Ga-HX01 was capable of clearly visualizing tumors, indicating its high sensitivity for tumor diagnosis.

To assess [^68^Ga]Ga-HX01’s performance across various tumor types, multiple tumor models were imaged using PET/CT. The results indicated that [^68^Ga]Ga-HX01 performed exceptionally well in visualizing tumors, thanks to its rapid clearance and low background uptake. Due to its low uptake in the brain and most abdominal organs, [^68^Ga]Ga-HX01 provided clear and distinct visualization of brain tumors against the surrounding normal brain tissue background. The tumor-to-brain ratio of [^68^Ga]Ga-HX01 (3.375 ± 0.261) was notably higher than that of [^18^F]FDG (1.120 ± 0.230, *p* < 0.05), indicating its potential for enhanced imaging of brain tumors and highlighting its utility in detecting and assessing these challenging malignancies. Additionally, the study compared [^68^Ga]Ga-HX01 with [^18^F]FDG, a widely used clinical PET imaging agent. The comparative analysis revealed that [^68^Ga]Ga-HX01 consistently provided better tumor contrast to background, higher tumor-to-muscle and tumor-to-blood ratios, and clearer tumor visualization, particularly in abdominal neoplasms.

[^68^Ga]Ga-HX01 PET/CT imaging exhibited great promise as a versatile and effective imaging agent for various tumor types such as glioma, head and neck squamous cell carcinoma, and esophageal cancer [44,45,46]. The comprehensive evaluation, including blocking studies, single target imaging, dynamic imaging, dose sensitivity assessments, and comparison with [^18^F]FDG, demonstrated its superior sensitivity, specificity, and imaging performance. These findings support the potential clinical utility of [^68^Ga]Ga-HX01 in diagnosing and monitoring different cancer types.

### 8.3. Pharmaceutical and Safety Assessments

[^68^Ga]Ga-HX01 consistently exhibited uptake patterns in accordance with the status of tumor neovascularization across various tumor types. The expression of integrin α_v_β_3_ and CD13 on tumor neovasculature made it an ideal target for broad-spectrum tumor imaging. Notably, [^68^Ga]Ga-HX01’s higher tumor uptake, compared to monomeric tracers, could be attributed to its enhanced avidity effect involving both RGD and NGR binding, which was further validated through the blocking studies. This agent also demonstrated rapid distribution and clearance, primarily via renal excretion, and low background accumulation in most organs, contributing to its superior tumor-to-background contrast. Importantly, the agent’s ability to deliver high-quality imaging at lower radioactivity doses suggests potential for reduced patient radiation exposure in clinical translation.

The outperformance of [^18^F]FDG in terms of pharmacokinetic properties, tumor-to-muscle, and tumor-to-blood ratios indicated its advantage in imaging tumors with low glucose metabolism and distinguishing between inflammation and malignancy. [^68^Ga]Ga-HX01 presents a promising imaging agent with excellent pharmaceutical characteristics. While there are some limitations to address, such as renal uptake, this study has paved the way for further clinical research and evaluation of [^68^Ga]Ga-HX01. It holds great potential for diagnostic imaging and molecular therapy, particularly for patients with challenging-to-diagnose tumors in the head and neck, and ongoing efforts are focused on its clinical translation.

## 9. DTPA-PEG-Fe_3_O_4_-RGD for the Purpose of SPECT/MRI [47]

### 9.1. Background and Production

This research focused on the development of a nanoprobe for SPECT/MR dual-modality imaging using magnetic FeNPs. These NPs were synthesized through pyrolysis and modified with 3-(3,4-Dihydroxyphenyl)propionic acid (DHCA). To enhance their functionality, RGDyk and DTPA ligands were grafted onto the NPs. RGDyk targets α_v_β_3_-rich tumor cells, facilitating their aggregation at tumor sites such as those found in the cancers of the head and neck [42]. DTPA was employed to label the radionuclide ^99m^Tc, enabling SPECT imaging. This dual-modality nanoprobe offered a powerful tool for tumor evaluation and diagnosis. Magnetic iron oxide NPs were chosen due to their biocompatibility and ability to accumulate in specific tissues, providing ^99m^T2 enhancement for MRI contrast [48]. By combining the strengths of SPECT’s high sensitivity with MRI’s high resolution, this approach aims to offer comprehensive and accurate information for disease diagnosis and treatment.

The researchers utilized various reagents and materials for the synthesis and characterization of magnetic iron oxide NPs modified for dual-modality imaging, as shown in Figure 5. Iron acetylacetonate, octadecene, oleic acid, oleylamine, and other chemicals were employed in their preparation. The synthesis involved the exchange of oleic acid ligands with DHCA to render the NPs hydrophilic. For functionalization, 1-ethyl-3-(3-dim-ethylaminopropyl) carbodiimide hydrochloride and N-hydroxysuccinimide (NHS) were used for covalent attachment of RGDyk and NH_2_-PEG_2000_-NH_2_ to the NPs, making them suitable for the targeting of α_v_β_3_-rich tumor cells. The NPs’ characteristics were analyzed through techniques such as TEM, XRD, ζ-potential, and DLS measurements. Additionally, the study investigated the magnetic properties and relaxation rates of the particles. Infrared spectra were obtained using FTIR, and contrast analysis was performed with a 1.5 T analyzer. 

### 9.2. Characteristics and Clinical Applications

The Fe_3_O_4_-DHCA NPs exhibited the general characteristics of having regular shape, monodispersity, and an average size of approximately 13 nm. These NPs maintained superparamagnetism at room temperature with a saturation magnetization of 71 emu/g. The surface modification with various organic compounds was confirmed through FTIR, showing the successful exchange of oleic acid with DHCA. DLS measurements yielded an average hydrodynamic diameter of 31 nm for the Fe_3_O_4_-DHCA NPs, which increased to 51 nm after grafting NH_2_-PEG_2000_-NH_2_ and RGDyk. The ζ-potential changed from −34.0 mV for the unfunctionalized NPs to −16.4 mV for the fully functionalized DTPA-PEG-Fe_3_O_4_-RGD NPs.

In terms of magnetic properties, the r_2_ relaxation rate of DTPA-PEG-Fe_3_O_4_-RGD NPs (181.446 mM^−1^ s^−1^) was notably higher than that of the commercial CA Feridex. MRI phantom imaging demonstrated the potential of DTPA-PEG-Fe_3_O_4_-RGD NPs as a t_2_ CA, as the t_2_-weighted phantom images darkened with increasing Fe concentration. These NPs were further radiolabeled with ^99m^Tc, resulting in a labeling yield of approximately 93%. The radiolabeling had minimal impact on the particle size and stability of DTPA-PEG-Fe_3_O_4_-RGD NPs.

In vivo studies on a 4T1 subcutaneous tumor model included SPECT and MR imaging. SPECT imaging revealed clear visualization of the tumor at 6 and 12 h post-injection, confirming the tumor-targeting properties. Furthermore, t_2_-weighted MR images indicated a significant decrease in signal intensity in the tumor region at 6 h post-injection, demonstrating the specific accumulation of DTPA-PEG-Fe_3_O_4_-RGD NPs in the tumor. Overall, this research successfully developed a dual-modality CA using magnetic iron oxide NPs, combining SPECT and MRI for the evaluation and diagnosis of malignant tumors.

### 9.3. Pharmaceutical and Safety Assessments

Biodistribution studies were conducted in 4T1 tumor-bearing mice to assess the tissue distribution of ^99m^Tc-DTPA-PEG-Fe_3_O_4_-RGD NPs. The results indicated rapid accumulation and good retention of the NPs in the tumor, peaking at 6 h post-injection (4.82 ± 1.63%ID/g). The tumor-to-muscle ratio at this time reached 5.17, highlighting the specificity of NP uptake in the tumor. In contrast, low activity was observed in the blood and muscle. The kidney showed an initial uptake of 29.21 ± 4.28%ID/g at 1 h, which decreased to 13.56 ± 2.88%ID/g after 12 h. The liver exhibited a peak uptake of 29.78 ± 1.64%ID/g at 1 h, decreasing to 16.95 ± 2.23%ID/g at 12 h, with excretion through the intestine. Similarly, the spleen showed a decrease in uptake from 8.80 ± 1.04%ID/g at 1 h to 2.27 ± 1.269%ID/g at 12 h. These trends were observed in other tissues and organs, indicating the gradual metabolism and elimination of ^99m^Tc-DTPA-PEG-Fe_3_O_4_-RGD NPs from the body.

Comparison with a previous study involving ^125^I-labeled dimeric cRGD peptide-modified iron oxide NPs showed similar tumor uptake but faster clearance from the kidneys and liver [49]. These variations could be attributed to differences in nuclide labeling methods and radio-metabolites. The results collectively demonstrated that ^99m^Tc-DTPA-PEG-Fe_3_O_4_-RGD NPs exhibit tumor-specific uptake, good retention, and the ability to be metabolized from the body over time.

## 10. Discussion

The landscape of head and neck cancer imaging is poised for a revolution with the advent of novel contrast agents, each designed to address specific limitations of traditional imaging modalities. The development of targeted imaging agents marks a significant shift towards precision medicine and can help to address some of the limitations of earlier contrast agents. By specifically binding to molecular markers, these agents have enhanced specificity and sensitivity, allowing for more accurate tumor delineation [12]. This advancement is critical, given the complexity of head and neck anatomy, where precise localization of tumors is paramount for effective treatment planning. Markers such as tumor hypoxia in the case of CAIX, which is often associated with poor prognosis and treatment resistance, can provide crucial insights into the tumor microenvironment [16]. However, the clinical translation of these agents requires addressing potential limitations, such as tumors in the head and neck region often exhibiting diverse biological characteristics, making it difficult to rely on a single marker for detection [14]. However, by creating agents that bind to several markers, researchers can potentially increase the likelihood of detecting tumors regardless of their specific molecular profile. Further research is also needed to validate their efficacy across a broader patient population and to ensure that they do not produce false negatives in tumors with low marker expression. 

Agents that exemplify the power of multimodal imaging, combining MRI with CT or CT with PET, offer a more holistic view of the tumor, integrating anatomical and functional information. This dual capability is especially valuable in head and neck cancers, where detailed imaging can significantly impact surgical planning and treatment monitoring [40]. The challenge lies in ensuring that the integration of these technologies does not complicate the imaging process or increase the risk of adverse effects. The synthesis and functionalization of these nanoparticles must be optimized to maintain their stability and biocompatibility in vivo as well.

Innovative approaches to enhancing MRI contrast, such as single-atom gadolinium and iron-based nanoparticles, represent significant advancements. These agents offer high relaxivity and rapid clearance, minimizing toxicity while providing superior imaging resolution [20]. The main challenge for these agents is their potential to elicit immune responses or accumulate in non-target tissues, leading to unforeseen side effects [20]. Rigorous testing and refinement of their chemical properties are essential to mitigate these risks.

Iodine-based activity probes and dual-targeting radioligands offer significant advancements in CT and PET/CT imaging. By targeting specific proteases or receptors, these agents provide high specificity and sensitivity, essential for early cancer detection and accurate staging [23]. However, the success of these agents depends on their ability to selectively target tumor tissues without affecting healthy cells. The risk of off-target effects and the potential for immunogenicity must be carefully evaluated. Additionally, the cost and complexity of synthesizing these agents may limit their widespread adoption.

The advancements discussed here have larger implications for the field of oncology. The shift towards multimodal and safer imaging agents with specific tissue target integration reflects a broader trend towards personalized medicine, where diagnostic and therapeutic strategies are tailored to individual patient profiles. These innovations could pave the way for more precise and effective cancer treatments, ultimately improving patient outcomes. Overall, three main challenges appeared across all agent types and tumor targeting strategies included in this review article. The variability in tumor biology, the potential for adverse effects, and the need for extensive validation studies. These issues are further exacerbated by the anatomical and physiological complexities of the head and neck region, potentially necessitating highly specialized solutions to these challenges. Collaborative efforts between researchers, clinicians, and regulatory bodies are essential to overcome these challenges and ensure the successful translation of these agents into clinical practice.

## 11. Conclusions

The field of diagnostic imaging for head and neck cancers is undergoing a transformative period with the advent of novel contrast agents that promise enhanced specificity, sensitivity, and overall imaging quality. This review has shed light on several innovative contrast agents that hold the potential to significantly improve the diagnosis, treatment planning, and monitoring of head and neck cancers. These innovations include their ability to target specific molecular markers associated with tumors. For instance, MT218 [ZD2-N3-Gd(HP-DO3A)] targets extradomain B fibronectin (EDB-FN), a marker overexpressed in aggressive tumors. This specificity allows for more precise tumor localization and characterization, leading to better treatment planning and monitoring for MRI. Similarly, AZA-TA-Mn targets carbonic anhydrase IX (CAIX), prevalent in hypoxic tumor regions. By providing detailed imaging of hypoxic areas, this agent helps in assessing tumor aggressiveness and potential treatment resistance, which are crucial for effective therapy planning. Alternatively, for CT imaging, IN-ABPs leverage activity-based probes to detect cathepsin activity within tumors, providing highly sensitive and specific cancer detection. These agents can significantly enhance early disease diagnosis, potentially leading to better patient outcomes.

The development of contrast agents with multimodal imaging capabilities, such as NaHoF_4_ nanoparticles and agents like [^68^Ga]Ga-HX01, represents a significant leap forward as well. NaHoF_4_ NPs offer dual-modality imaging for both CT and MRI, combining the anatomical detail of CT with the superior soft tissue contrast of MRI. This dual capability provides comprehensive diagnostic information in a single session, streamlining the diagnostic process and improving accuracy. [^68^Ga]Ga-HX01, with their dual PET and CT imaging capabilities, ensure precise tumor delineation, particularly in complex anatomical regions like the head and neck. These agents enhance both preoperative planning and postoperative monitoring, reducing the risk of recurrence and improving surgical outcomes.

A common theme among the novel contrast agents is their improved safety profile compared to traditional agents. For example, Gd-SA, designed with single-atom gadolinium, offers high relaxivity with reduced toxicity. Its ability to clear rapidly from non-target tissues minimizes potential side effects, making it a safer alternative for patients. Iron-based nanoparticles, such as DTPA-PEG-Fe_3_0_4_-RGD, are another example, offering high biocompatibility and stability. These agents provide effective tumor visualization with minimal toxicity, making them suitable for repeated use in monitoring treatment progress.

Overall, the integration of these innovative contrast agents into clinical practice promises to elevate the standards of head and neck cancer imaging. Enhanced specificity and sensitivity, multimodal capabilities, reduced toxicity, and improved biocompatibility will collectively contribute to more accurate diagnoses, effective treatment planning, and comprehensive monitoring of therapeutic outcomes. The advancements highlighted in this review underscore the importance of continued research and development in this field. As some of these agents potentially progress through clinical trials and into clinical use, they hold the potential to significantly improve the quality of life and survival rates for patients with head and neck cancers. The future of radiological imaging in oncology looks promising, with these novel agents spearheading the way for more personalized and precise medical care.

## Figures and Tables

**Figure 1 diagnostics-14-01666-f001:**
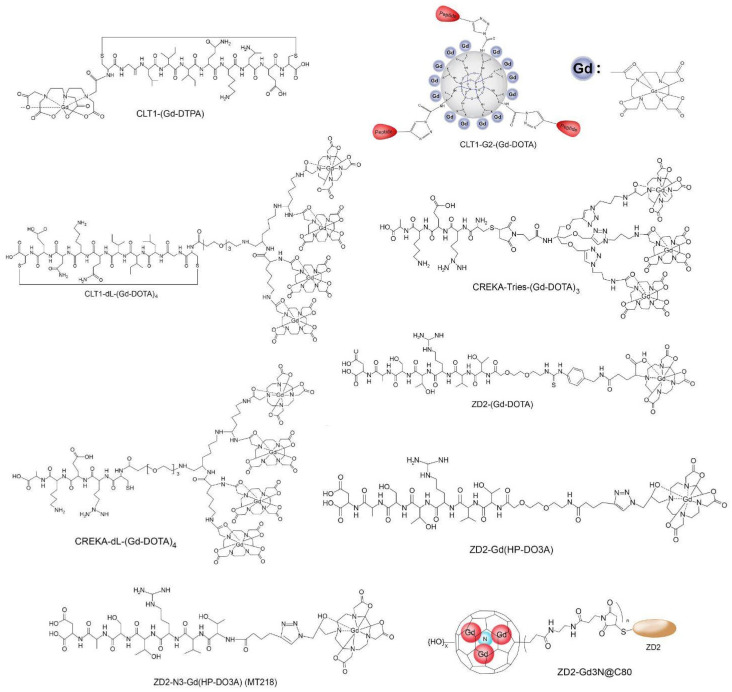
Chemical structures of GBCAs specific to EDB or fibrin–fibronectin.

**Figure 2 diagnostics-14-01666-f002:**
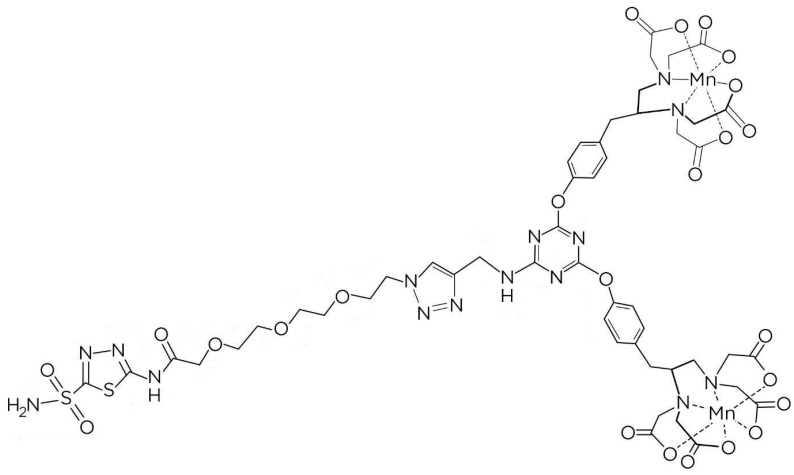
Chemical structure of AZA-TA-Mn complex.

**Figure 3 diagnostics-14-01666-f003:**
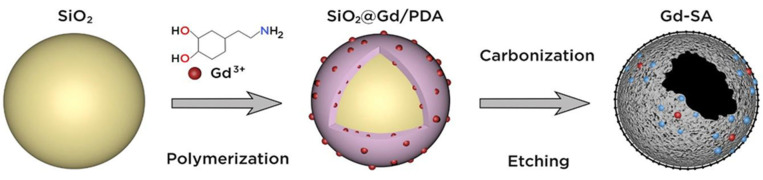
Synthesis protocol and morphology of Gd-SA.

**Figure 4 diagnostics-14-01666-f004:**
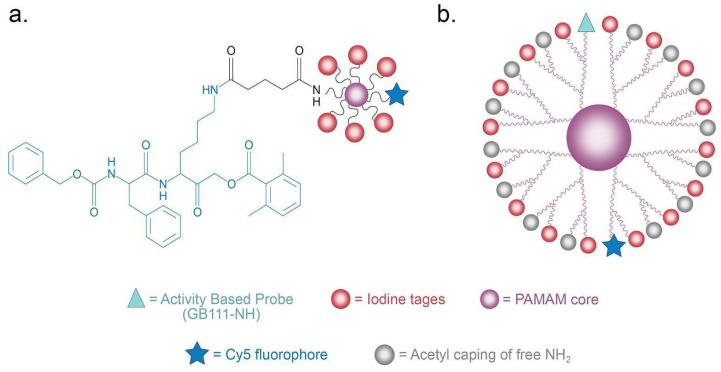
Structure of G1 IN-ABPs (**a**) and PAMAM G3 IN-ABPs (**b**).

**Figure 5 diagnostics-14-01666-f005:**
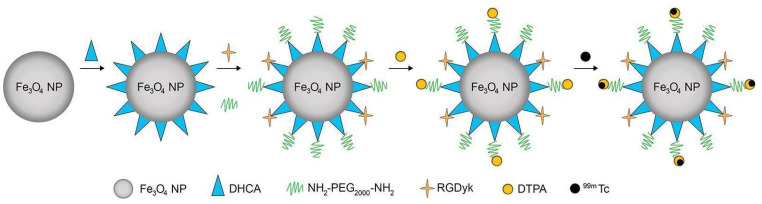
Synthesis schematic of DTPA-PEG-Fe304-RGD.

**Table 1 diagnostics-14-01666-t001:** Agents highlighted in this review article which are sorted by their appearance within the text.

Agent	Type	Target	Imaging Modality	Clinical/Preclinical Stage
MT218	Peptide	EDB-FN	MRI	Preclinical
AZA-TA-Mn	Small molecule	CAIX	MRI	Preclinical
Biotin/PEG-UCNPs	Nanoparticle	Biotin receptors	MRI	Preclinical
Gd-SA	Nanoparticle	Tumor tissues	MRI	Preclinical
IN-ABPs	Nanoparticle	Cathepsin proteases	CT	Preclinical
NaHoF_4_	Nanoparticle	Tumor tissues	MRI/CT	Preclinical
[^68^Ga]Ga-HX01	Peptide	Integrin α_v_β_3_ and CD13	PET/CT	Phase I clinical
DTPA-PEG-Fe_3_O_4_-RGD	Nanoparticle	Integrin α_v_β_3_	SPECT/MRI	Preclinical

## Data Availability

No new data were created or analyzed in this study. Data sharing is not applicable to this article.

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
