# Peer review of "Emerging Head and Neck Tumor Targeting Contrast Agents for the Purpose of CT, MRI, and Multimodal Diagnostic Imaging: A Molecular Review"

_diagnostics, 2024, doi:10.3390/diagnostics14151666_

Round 1

Reviewer 1 Report

Comments and Suggestions for Authors

The review summarized newly published head and neck tumor targeting MR and CT contrast agents. It is an interesting topic. But unclear organization through the manuscript. And these contrast agents are tumor targeting, not special for Head and neck tumor. The review needs a major revision, and resubmission after major revision.

Comments:

1.       The structure of the manuscript needs to be changed.

(1)Tumor targeting MR CAs including MT218, AZA-TA-Mn, KMnF3/Yb/Er, Gd-SA;

(2)Tumor targeting CT In-ABPs;

(3)Bimodal CT/MRI NaHoF4, SPECT/MRI DTPA-PEG-Fe3O4-RGD, [68Ga]Ga-HX01 PET/CT

2.       Delete “1.1 search strategy” in the introduction, this is the method for search all references.

3.       In 2. MT218, 2.1 Background and production, author only quoted the Figure 1, but no production is introduced or discussed.

4.       In 3. AZA-TA-Mn section,

5.       Where are the compounds 1-4? Only see Figure 2.

6.       Before the figure 4, no figure 3, and there are two Figure 5!

7.       In 10, discussion section is a summary, not a discussion. Indeed, authors need to discuss why these contrast agents are for HEAD and Neck tumor targeting.

8.       All the figures need to get permission for publishing.

9.       No reference number quoted in the section title. Such as MT218[ZD3-N3-Gd(HP-DO3A)] for the purpose of MRI [9]. It is confused.

Comments on the Quality of English Language

Minor editing of English language required

Reviewer 2 Report

Comments and Suggestions for Authors

The Review article "Emerging Head and Neck Tumor Targeting Contrast Agents for the Purpose of CT, MRI, and Multimodal Diagnostic Imaging: A Molecular Review" represented an overall outlook of the imaging tools available for head and neck tumors. The article can be accepted after some minor modifications suggested below:

(1) The introduction mostly focused on nanoparticles and less on small molecules/peptides. Peptides are more promising than nanoparticles for clinical translation. Please incorporate more information on peptides in the introduction. 

(2) I recommend that the author incorporate a systematic table for all the agents described in the review with clinical/preclinical staging.

(3) It would be beneficial to include the current status of antibody and protein-based agents for head and neck tumors. This will keep the reader informed and up-to-date with the latest developments in the field.

(4) I recommend the inclusion of other CAIX targeting ligands in section 3. This will provide a more comprehensive and thorough understanding of the topic for the reader.

(5)  Please incorporate the statistical comparison in lines 394-395.

(6) In section 8, the authors incorporated PET/CT imaging probes without any prior background in the Abstract and Introduction. Emerging PET/CT imaging agents for Head and Neck Tumor could be a separate literate search/review article, which is unnecessary to incorporate here.

(7) The Conclusion section needs to be improved based on the information given in the main text. 

Round 2

Reviewer 1 Report

Comments and Suggestions for Authors

After the major revision, the manuscript is now ready to be accepted in its current form.